# Steroid Pulse Therapy Leads to Secondary Infections and Poor Outcomes in Patients with Severe Acute Respiratory Syndrome Coronavirus 2 (SARS-CoV-2) in Intensive Care Units: A Retrospective Cohort Study

**DOI:** 10.3390/v17060822

**Published:** 2025-06-06

**Authors:** Katsuhiro Nakagawa, Shingo Ihara, Junko Yamaguchi, Tsukasa Kuwana, Kosaku Kinoshita

**Affiliations:** Division of Emergency and Critical Care Medicine, Department of Acute Medicine, Nihon University School of Medicine, 30-1 Oyaguchi Kamimachi, Itabashi-ku, Tokyo 173-8610, Japan; nakagawa.katsuhiro@nihon-u.ac.jp (K.N.); ihara.shingo@nihon-u.ac.jp (S.I.); kuwana.tsukasa@nihon-u.ac.jp (T.K.); kinoshita.kosaku@nihon-u.ac.jp (K.K.)

**Keywords:** COVID-19, pneumonia, steroid pulse therapy, remdesivir, secondary infection

## Abstract

The efficacy of steroid pulse therapy for treating severe coronavirus disease (COVID-19) pneumonia remains unclear. This study aimed to determine the efficacy of steroid pulse therapy for severe COVID-19 pneumonia in patients who did not respond to conventional therapy, including steroids. We included 76 patients with severe COVID-19 pneumonia treated with steroids in this single-facility retrospective observational study. Severe COVID-19 pneumonia was defined as requiring high-concentration oxygen administration (oxygen mask with reservoir mask (RM) > 6 L/min), high-flow nasal cannula oxygen therapy, or ventilatory support for respiratory control. The patient characteristics at admission and changes in them over time were examined in (a) a survival vs. death group, and (b) a steroid pulse vs. non-steroid pulse therapy group. Steroid pulse therapy significantly improved the ratio of partial pressure of arterial oxygen to fraction of inspired oxygen just after the therapy and after one week of therapy, but had no effect on the sequential organ failure assessment scores over time. Multivariate logistic regression analyses showed that remdesivir use was associated with better survival outcomes, while steroid pulse therapy was associated with poor outcomes. In conclusion, steroid pulse therapy did not improve the prognosis of patients with severe COVID-19 pneumonia any more effectively than conventional steroid therapy.

## 1. Introduction

Although the number of deaths related to coronavirus disease (COVID-19) after acquiring immunity decreased significantly from the start of the pandemic until 10 November 2024, over 776.8 million confirmed COVID-19 cases and over 7 million confirmed deaths were reported to the WHO across 234 countries [1]. Steroid therapy has been reported to improve outcomes in patients with COVID-19 pneumonia requiring oxygen therapy [2], but the optimal dose of steroids remains unknown [3].

While some studies have suggested the potential for clinical improvement with steroid therapy in patients with COVID-19 pneumonia [4,5], other reports state that steroid therapy does not effectively prevent an excessive immune response from the host, resulting in poor outcomes in cases of severe COVID-19 pneumonia [6,7]. Thus, the responsiveness to steroid treatment in COVID-19 pneumonia remains controversial. These discrepancies may, in part, be attributed to differences in disease severity among the study populations and variations in the administered steroid dosages. In particular, the effectiveness of steroid pulse therapy in patients with severe COVID-19 pneumonia who do not respond to conventional treatment remains unclear. Therefore, the aim of this study was to evaluate the efficacy of steroid pulse therapy in patients with severe COVID-19 pneumonia refractory to standard care.

## 2. Materials and Methods

This single-institution, retrospective observational study used the database of patients who were diagnosed with COVID-19 pneumonia at our hospital. The study was approved by the Clinical Research Review Committee of Nihon Itabashi University Hospital (RK-220809-8). The requirement for informed consent was waived by the approving authorities owing to the retrospective nature of this study. We included patients who (a) were aged 20 years or older, (b) were admitted to the ICU of our hospital between 1 April 2020 and 31 October 2021, and (c) required high-concentration oxygen therapy (oxygen mask with reservoir > 6 L/min), high-flow nasal cannula oxygen therapy (HFNCO), or ventilatory management, which meet the criteria for severe COVID-19 as per the Japanese guidelines [8]. COVID-19 pneumonia was diagnosed based on a positive severe acute respiratory syndrome coronavirus 2 (SARS-CoV-2) polymerase chain reaction (PCR) response. Patients who (a) had not received any steroid therapy or (b) did not belong to the severe category were excluded from the study. Steroid pulse therapy is typically administered for three days, after which the effects become noticeable. Hence, in patients who died early (early death was defined as death within four days of entering the ICU or within one day of receiving steroid pulse therapy), the effects of the therapy could not be assessed, and they were therefore excluded from the study. We also excluded patients who developed a bacterial infection at the time of admission or before receiving steroid pulse therapy, or were being treated for any infection other than COVID-19 pneumonia prior to steroid pulse therapy, because we believe that this would have modified the benefit or disadvantage of steroid administration. The flowchart in Figure 1 summarizes the selection of study participants. All patient data, including vital signs, clinical and laboratory data, and imaging findings, were obtained from hospital databases and patients’ clinical records. Each of these parameters was associated with the severity of SARS-CoV-2 pneumonia and comorbidities, nutritional status at the time of admission, the characteristics of the treatment methods, and chest X-ray findings, and hence, each parameter was a predictor of patient outcomes.

The effect of steroid pulse therapy was examined by calculating the sequential organ failure assessment (SOFA) score [9], which indicates the severity of organ failure. The decision to initiate steroid pulse therapy was made at the discretion of the attending physician and was applied to patients who met at least one of the following criteria: (1) a ratio of partial pressure of arterial oxygen to fraction of inspired oxygen (P/F) of less than 200 at the time of admission; (2) progressive deterioration in oxygenation status despite the initiation of standard treatment for COVID-19 pneumonia; (3) clinical worsening in oxygenation status from the time of admission, despite receiving standard therapy (e.g., escalation from oxygen via reservoir mask to mechanical ventilation); and (4) radiological deterioration observed on chest X-ray scans, such as the expansion of pulmonary infiltrates or increased ground-glass opacities (GGOs).

Patients were divided into two groups based on whether they received steroid pulse therapy or not. 

### 2.1. Steroid Pulse Therapy for Severe COVID-19

While several studies have explored effective treatments for COVID-19, the treatment guidelines published by the National Institutes of Health [5] currently recommend dexamethasone and remdesivir as standard therapy. Steroid pulse therapy is a short-term, high-dose intensive treatment expected to have strong anti-inflammatory and immunosuppressive effects [4]. It is used for severe conditions, including severe COVID-19 pneumonia, that do not respond to normal steroid doses. However, the criteria and methods for administering steroid pulse therapy for COVID-19 pneumonia still have not been defined worldwide [3].

Steroid pulse therapy using methylprednisolone consists of the intravenous administration of 10 to 20 mg per kg of body weight (1000 mg of methylprednisolone/50 kg of body weight) over 30 min to one hour for one to five days, every day or every other day [3]. In Japan, steroid pulse therapy generally consists of the administration of 1000 mg/day of methylprednisolone for three days, followed by post-treatment with prednisone in a tapered dose according to standard body weight [4,10]. In this study, 1000 mg/day of methylprednisolone was administered daily for three days, followed by 1 mg/kg/day of prednisone for another three days, and then, the dosage was tapered with a 5 to 10 mg reduction every three days to complete the steroid pulse therapy. Based on previous reports, we used the P/F ratio < 200 criterion for increasing the dosage of steroid in patients with severe COVID-19 pneumonia [11,12].

### 2.2. Criteria for Non-Response to Conventional Therapy

Patients with a P/F ratio < 200 and those who deteriorated in response to the oxygen therapy at the time of admission (e.g., patients who went from oxygen therapy with an oxygen mask with reservoir mask to requiring ventilator management) were considered non-responsive to conventional therapy based on previous reports [11,12].

Patients with infiltrative shadows or an enlarged ground-glass opacity (GGO) on imaging studies were also considered non-responsive to conventional therapy. This is because there is no standard for the quantification of infiltrating shadows and an enlarged GGO on imaging studies in COVID-19 pneumonia evaluation. While previous studies have primarily evaluated pulmonary disease progression using computed tomography (CT), a distinctive feature of our study is that the patients included were critically ill, and a follow-up with chest CT was not always feasible. Therefore, we assessed disease progression using plain chest radiographs.

Although there are no objective radiographic indicators specifically validated for COVID-19 pneumonia using plain chest radiography, we re-evaluated the radiographic findings of our patient cohort using the RALE (Radiographic Assessment of Lung Edema) score, which is based solely on plain chest X-ray findings [13]. The RALE score is a semi-quantitative scoring system developed to assess the extent and density of pulmonary edema on chest diagraphs in patients with acute respiratory distress syndrome (ARDS).

In this system, the chest radiograph is divided into four quadrants. For each quadrant, the extent of alveolar opacities is scored from 0 to 4, and the density of consolidation is scored from 1 to 3. The product of these two scores is calculated for each quadrant, and the sum of the scores across all quadrants yields a total RALE score ranging from 0 to a maximum of 48. The RALE score has been shown to correlate with oxygenation status and clinical outcomes in patients with ARDS, suggesting its clinical utility in monitoring treatment response and predicting prognosis.

### 2.3. Secondary Infections

Secondary infections refer to all bacterial and fungal infections that occur after the start of steroid therapy. We examined the occurrence of secondary infections to assess the prognosis following the steroid therapy. Specimens from sites of possible infection based on clinical findings as well as imaging and laboratory data were collected and used for bacterial culture. The cause of infection was determined by combining the clinical, imaging, laboratory, and bacterial culture findings [14]. Specimens consisted of blood, sputum, or urine samples, depending on the individual case.

### 2.4. Ventilator-Free Days (VFD)

VFD was defined as the total number of days that patients were weaned from the ventilator during the 28-day observation period after being placed on the ventilator [15]. However, VFD did not include the number of days for patients who died during the observation period, even if they were extubated, or for patients who were reintubated within 48 h after withdrawal.

### 2.5. CONUT Score

Undernutrition is a significant problem in clinical situations. CONUT is a clear, accurate, easy-to-use screening tool for undernutrition. Based on serum albumin levels, peripheral blood lymphocyte counts, and total cholesterol levels [16], the CONUT score is used for the early detection and management of malnutrition in hospitals (Appendix A).

### 2.6. Statistical Analysis

All statistical analyses were performed using SPSS version 29 (IBM Corp., Armonk, NY, USA) and JMP ver. 14.2 (SAS Institute, Cary, NC, USA). The data are presented as the mean values (standard deviation [SD]), median values (interquartile range), or number of cases (%). Statistical significance was set at *p* < 0.05. Continuous variables were compared using Student’s *t*-test or a Mann–Whitney U test and Wilcoxon’s signed-rank test, as appropriate. A Chi-square or Fisher’s exact probability test was performed for categorical variables. Multiple logistic regression analysis was used to predict prognostic factors in patients with severe COVID-19 pneumonia. Variables with *p*-values < 0.2 in the bivariate models (age, smoking, favipiravir, remdesivir, tocilizumab, steroid pulse therapy, lactate, and ferritin) were included in the multivariate model. We used multiple logistic regression analysis and variate models (age, BMI, smoking, LDH, ferritin, KL-6, WBC, lactate, favipiravir, remdesivir, and duration from onset to hospitalization) to predict prognostic factors in patients on steroid pulse therapy. Statistical significance was set at *p* < 0.05.

## 3. Results

### 3.1. Patient Characteristics

Table 1 compares the characteristics of patients in the two groups at admission.

The median age of all patients was 61.0 years (range: 54.8–75.0 years), 58/76 (76.3%) were men, and the median body mass index (BMI) was 25.7 kg/m^2^ (range: 22.2–27.7 kg/m^2^). A total of 28 patients (36.8%) had a history of diabetes mellitus, 6 (7.9%) of malignancy, 37 (48.7%) of hypertension, and 2 (2.6%) of chronic kidney disease. Forty patients (52.6%) smoked regularly.

Patients were divided into steroid pulse (*n* = 45, 59.2%) and non-steroid pulse (*n* = 31) therapy groups. The vital signs and comorbidities related to severe COVID-19 pneumonia prognosis were comparable between the two groups. However, compared to the non-steroid pulse therapy group, patients in the steroid pulse therapy group had significantly higher BMIs (*p* = 0.0430), lower lymphocyte counts (*p* = 0.045), and lower neutrophils counts (*p* = 0.021; Table 1A). Based on the CONUT scores, all patients were undernourished, with no significant differences between the two groups (Table 1B). The number of patients who had received remdesivir and baricitinib treatment was significantly higher in the steroid pulse therapy group (Table 1C), while VFD was significantly higher in the non-steroid pulse therapy group (*p* = 0.0295; Table 1D).

Table 2 shows the results of the multiple logistic regression analysis of the prognostic factors for all patients. Remdesivir therapy emerged as a significant prognostic factor (odds ratio [OR], 8.202; 95% confidence interval [CI], 1.479–49,495; *p* = 0.008), while steroid pulse therapy was associated with poor outcomes (OR 0.032; 95% CI, 0.004–0.240; *p* < 0.001) (Table 2).The predictive model equation was ([Age] × 0.920 + [received remdesivir treatment] × 8.202 − 0.032 × [steroid pulse therapy] + 7.204). Using the values derived from this equation, the predicted probabilities for each case were calculated and evaluated using a receiver operating characteristic (ROC) curve. The ROC analysis demonstrated an area under the curve indicating good predictive performance of the model (AUC = 0.847, 95% CI: 0.748–0.945, *p* < 0.0001) (Appendix A).

Multivariate logistic regression analyses (stepwise increase in the number of variables) were conducted with the explanatory variables age, smoking, favipiravir, remdesivir, tocilizumab, steroid pulse therapy, lactate, and ferritin.

We used multiple logistic regression analysis to determine prognostic factors in patients receiving steroid pulse therapy (variates included age, BMI, smoking, LDH, ferritin, KL-6, WBC, lactate, favipiravir, remdesivir, and duration from onset to hospitalization). Age and remdesivir therapy emerged as independent factors associated with better outcomes following steroid pulse therapy (Table 3).

The incidence of secondary infections was significantly higher among non-survivors (Appendix A). Compared to the non-steroid therapy group, the steroid pulse therapy group had a higher incidence of infectious pneumonia (other than COVID-19 pneumonia; 66.7% vs. 48.4%, *p* = 0.1110), urinary tract infection (26.7% vs. 25.8%, *p* = 0.9333), and bacteremia (31.1% vs. 22.6%, *p* = 0.4138), and included overlapping cases (Figure 2). The overall incidence of secondary infections was not significantly different between the two groups (*p* = 0.0598, Table 1E). These findings show that the increase in the total dose and duration of steroid administration had no significant effects on the incidence of secondary infections (Appendix A).

### 3.2. Changes in Patient Parameters over Time

Appendix A show the changes in the parameters over time (at admission, before therapy, and after therapy). In the steroid pulse therapy group, parameters were compared at admission vs. after one week of therapy, and at four days after admission vs. 11 days after admission.

The P/F ratio decreased significantly (*p* = 0.0494) in the steroid pulse therapy group from admission (median: 200.0, IQR: 76.0–316.7) to just before therapy (median: 124.0, IQR: 89.0–163.4). It also showed a significant improvement from before therapy to immediately after therapy (median: 157.0, IQR: 113.5–203.5) and one week after therapy (median: 190.0, IQR: 125–273.3, *p* = 0. 0175, *p* = 0.0014). In contrast, in the non-steroid pulse therapy group, the P/F ratio increased significantly from admission (median: 130.0, interquartile range: 97.0–220.0) to four days after admission (median: 205.0, IQR: 149.0–280.0) and 11 days after admission (median: 303.3, IQR: 126.0–457.1, *p* = 0.0014. *p* = 0.0106, *p* = 0.0017). The P/F ratio also increased significantly from four to 11 days after admission (*p* = 0.0398; Appendix A).

Specifically, there was no significant difference in the SOFA scores between the steroid pulse therapy and non-steroid pulse therapy groups at admission [4.0 (3–5) vs. 4.0 (3.0–6.5); *p* = 0.416] (Table 1B). However, in the steroid pulse therapy group, the SOFA scores remained unchanged over time—[4.0 (4–6)] at admission, [4.0 (4–6)] on day 4 after treatment, and [4.0 (2–7)] on day 11 (*p* = 0.2964). In contrast, the non-steroid pulse therapy group showed significant improvement in SOFA scores from [4.0 (3–7)] at admission to [4.0 (2–4)] on day 4 and [2.0 (1–4)] on day 11 (corresponding to one week after steroid pulse therapy initiation in the other group) (*p* = 0.0013) (Appendix A).

## 4. Discussion

In this study, steroid pulse therapy improved the P/F ratios in patients with severe novel COVID-19 infection who did not respond to conventional therapy (Appendix A). However, it did not improve their overall outcome. Logistic regression analysis demonstrated that steroid pulse therapy was an independent factor associated with poor outcomes (Table 2). While all patients died of respiratory failure from complications of severe COVID-19 pneumonia or additional bacterial pneumonia, steroid pulse therapy increased the incidence of secondary infections in severe patients in the ICU (Figure 2). The benefits of steroid pulse therapy for the treatment of COVID-19 pneumonia have been controversial due to differences in the types, doses, and duration of steroid administration, as well as the severity of the patients in each study.

### 4.1. Pros and Cons of Steroid Pulse Therapy

Excessive host immune response from previous viral pneumonia (H5N1 influenza, severe acute respiratory syndrome, or H1N1 influenza) has been suggested to be a mechanism underlying severe COVID-19 infection, leading to organ damage and the development of acute respiratory distress syndrome (ARDS) [17]. Steroids are used for treating COVID-19 pneumonia because of their ability to mitigate immune responses and excessive inappropriate inflammation [18]. Steroid therapy, including pulse therapy, is known to reduce ARDS-related mortality in patients with COVID-19 novel coronavirus infection [19] and has been used in Japan [5,20]. Dexamethasone has been shown to improve survival outcomes in patients with severe COVID-19 [2].

In addition, glucocorticoids such as methylprednisolone have been shown to block excessive inflammatory responses and death in patients with COVID-19 by inhibiting the migration of neutrophils and monocytes to the inflammatory site [21]. Edalatifard et al. reported that the treatment of COVID-19 pneumonia patients without ARDS with methylprednisolone 250 mg/day for 3 days in addition to conventional therapy decreased mortality and improved lung damage, oxygen saturation level, and inflammatory markers [22]. In Japan, the administration of methylprednisolone pulse therapy to patients with severe coronavirus pneumonia on ventilators has been shown to reduce the risk of in-hospital mortality [4]. Contrarily, other reports have indicated that steroid therapy may fail to adequately suppress excessive host immune responses and may even be associated with poor outcomes in cases of severe COVID-19 pneumonia [7].

Therefore, the effectiveness of steroid therapy in the treatment of COVID-19 pneumonia remains a matter of ongoing debate.

Consistent with previous studies, our results show a time-dependent improvement in the P/F ratio, which was included in the ARDS diagnostic criteria in the steroid pulse therapy group (Appendix A). Our results also confirm the findings of the only reported randomized clinical trial (RCT) [6], which showed that steroid pulse therapy was ineffective. The RCT included 304 patients with moderate or severe COVID-19 pneumonia. There was no reduction in the length of hospital stay until discharge with steroid pulse therapy (1000 mg/day of methylprednisolone) for 3 days [6]. Furthermore, there was no significant difference in the ICU admission rates or mortality rates compared to the non-steroid pulse group. The target population of this trial differed from that of our study in that it did not include intubated patients or patients with multiple-organ failure who were admitted to the ICU. Yet, our findings were consistent with those from the trial, although we included patients who required ventilator support and those who developed multiple-organ failure. It has been suggested that SARS-CoV-2 may directly disrupt the normal function of the kidneys, liver, and peripheral blood components, increasing the risk of multiple-organ failure [23]. While steroids are known to alleviate excessive and inappropriate inflammation and immune responses [18], it is unclear whether they can improve the outcomes of all patients with coronavirus pneumonia [24].

The increase in secondary infections in response to steroid pulse therapy might contribute to its unfavorable outcomes. A study of 150 patients from two hospitals in Wuhan, China, reported that 11 patients out of 68 (16%) among those who died, and 1 patient out of 82 (1%) among those who were discharged, had a secondary infection [25]. Severe COVID-19 pneumonia is prone to a combination of secondary infections such as ventilator-associated pneumonia, bacteremia, and fungal infections because of its clinical features and the use of immunosuppressive drugs for treatment [26,27,28]. In this study, the incidence of secondary infections in the steroid pulse therapy group was higher than that in the non-treatment group, although the difference was not significant (Table 1).

In our study, the incidence of secondary infections was significantly higher in the non-survival group than in the survival group (Appendix A). In addition, multivariate analysis of patients who received steroid pulse therapy showed that older age was an independent factor for poor prognosis (Table 3). This is consistent with the findings of another study that compared patients with COVID-19 pneumonia who did and did not receive steroid pulse therapy and found that patients who were in their late 70s and older did not respond to treatment due to serious complications [29]. The occurrence of secondary infections was one of the causes of poor outcomes in these patients. In addition, patients admitted to the ICU and treated with corticosteroids, or who are in an immunosuppressed state or on ventilator support, are known to be susceptible to bacteremia, including infection with *Staphylococcus aureus* [30]. All patients in our study died due to respiratory failure associated with severe COVID-19 pneumonia or additional complications from bacterial pneumonia. In previous studies in which steroid pulse therapy was reported to be effective, the severity of the disease was lower than that in this study, and there were fewer cases of secondary infection [5,22]. Pappas et al. investigated lymphocyte dynamics in patients with severe COVID-19 pneumonia who had received dexamethasone therapy. Their findings indicated that alterations in peripheral blood lymphocyte subsets were associated with disease severity and clinical outcomes, regardless of dexamethasone administration. These results highlight the heterogeneity of host immune responses among patients with COVID-19 and suggest that the therapeutic efficacy of steroid treatment may vary depending on individual lymphocyte dynamics [31]. This may partly explain both the previously reported controversial effects of steroid pulse therapy and our own findings suggesting an increased incidence of secondary infections.

### 4.2. Multiorgan Failure

Patients in the ICU receiving steroid pulse therapy showed progressive multiorgan failure (Appendix A). In these patients, SARS-CoV-2 may directly control the normal function of the kidneys, liver, and peripheral blood components, increasing the risk of exacerbation of multiple-organ failure [23]. It remains unclear whether the outcomes can be improved by controlling only the excessive immune response.

### 4.3. Steroid Dosage

Several studies have been conducted on the treatment of COVID-19 pneumonia with methylprednisolone (250–500 mg/day) for three days [32]. Another reason steroid pulse therapy did not improve the prognosis of patients with severe COVID-19 pneumonia in our study, unlike in some other studies [4], may be the difference in the presence or absence of standard steroid therapy. In a previous study from Japan [4], only 14.4% of patients with severe COVID-19 pneumonia who required intubation received standard steroid therapy (6 mg/day of dexamethasone) before steroid pulse therapy. In contrast, 85.7% of patients who required intubation received prior standard steroid therapy in our study (Table 1C). In addition, Moromizato et al. reported that the risk of in-hospital death decreases when the interval between intubation and the start of steroid pulse therapy is short (less than five days). In our study, a significantly higher mortality rate was observed among patients who needed intubation compared to those who did not (64.3% vs. 5.6%, *p* = 0.0004). However, a sub analysis that included only patients with a short interval between intubation and the start of steroid pulse therapy (less than five days) showed no significant difference in outcomes between the two groups in our study (steroid pulse therapy vs. no steroid pulse therapy: 60.0% vs. 23.1%, *p* = 0.0721). While the appropriate timing of steroid pulse therapy remains unclear [33,34], these results suggest that it might be necessary to strengthen treatment from standard steroid therapy to steroid pulse therapy earlier for patients with the most severe form of the disease and who require intubation.

### 4.4. Duration of Treatment

All deaths registered during our study resulted from respiratory failure due to complications of severe pneumonia or additional bacterial pneumonia. The reason for the high incidence of secondary infections in the steroid pulse therapy group may be the increased steroid dose and prolonged steroid administration period due to the prednisolone post-treatment [35].

No significant differences were found between patients with and without secondary infections in terms of total steroid dose or administration period during treatment (Appendix A).

### 4.5. Remdesivir Treatment

Consistently with previous studies [36,37], remdesivir treatment improved prognoses (Table 3). However, patients in the steroid pulse therapy group had poor outcomes despite receiving significant amounts of remdesivir (*p* = 0.038). In other words, while remdesivir had the potential to improve outcomes, its effect was limited in patients receiving steroid pulse therapy.

### 4.6. Study Limitations

This study has some limitations. First, this study was a single-center, retrospective analysis with a relatively small sample size, which may have limited its statistical power and made it difficult to fully eliminate the influence of random effects. To assess multicollinearity among the independent variables (Table 2), the variance inflation factor (VIF) was calculated for each predictor [38].

The variance inflation factors (VIFs) for age, remdesivir treatment, and steroid pulse therapy were 1.117, 1.143, and 1.077, respectively. The tolerance values were also within acceptable ranges (Appendix A).

As none of the variables exceeded the commonly accepted thresholds indicative of strong multicollinearity (i.e., VIF > 10 or tolerance < 0.1), the likelihood of multicollinearity was considered low based on these cutoff values. However, as noted by O’Brien, reliance solely on VIF values for assessing multicollinearity is not sufficient. He emphasizes that the interpretation of multicollinearity should take into account the purpose of the model (e.g., prediction vs. causal inference), the importance of each variable, and the relationships among the predictors [39]. Thus, we divided the overall dataset into a development cohort and a validation cohort and confirmed the predictive performance of the model (Appendix A). The performance of the prediction model was evaluated using ROC analysis. The area under the ROC curve (AUC) was 0.880 (95% CI: 0.728–0.94, *p* = 0.004), indicating good discriminatory ability (Appendix A). Second, the characteristics of the study population and the treatment protocols were specific to our institution. Although the indication for steroid pulse therapy was determined based on multiple clinical considerations in accordance with national guidelines in Japan, at the time of treatment, there was still a lack of established evidence regarding the optimal dosage and timing of steroid administration for the patient population in this study. Therefore, our findings should be interpreted with caution, and their external validity (generalizability) to other institutions or regions requires further investigation. Third, in this study, steroid pulse therapy was administered to patients who did not respond to initial treatments (82.2% received initial standard steroid therapy), suggesting that the severity of COVID-19 pneumonia in the steroid pulse group may have been inherently higher, potentially contributing to poorer clinical outcomes. As this study focused on evaluating the effectiveness of steroid pulse therapy, it was not possible to fully investigate other background factors that may have influenced the observed outcomes. A significant decrease in lymphocyte count was observed in the steroid pulse therapy group. This finding is consistent with those in previous reports indicating that lymphopenia is a characteristic feature of severe COVID-19 pneumonia [40]. However, we did not examine lymphocyte subsets in this analysis. To better understand the effects of steroid pulse therapy, further investigation focusing on the immune response in COVID-19 patients is warranted to elucidate the underlying pathophysiology. To assess the reproducibility and generalizability of our findings, further validation through multicenter collaborative studies and prospective trials is warranted.

## 5. Conclusions

Our findings show that steroid pulse therapy does not improve the prognosis of patients with severe COVID-19 pneumonia over conventional steroid therapy. While all patients died of respiratory failure from complications of severe COVID-19 pneumonia or additional bacterial pneumonia, steroid pulse therapy increased the incidence of secondary infections in severe patients in the ICU.

Further research is needed to investigate the relationship between steroid pulse therapy and the occurrence of secondary infections in patients with different types of coronavirus pneumonia, and to evaluate the therapeutic efficacy of this intervention.

## Figures and Tables

**Figure 1 viruses-17-00822-f001:**
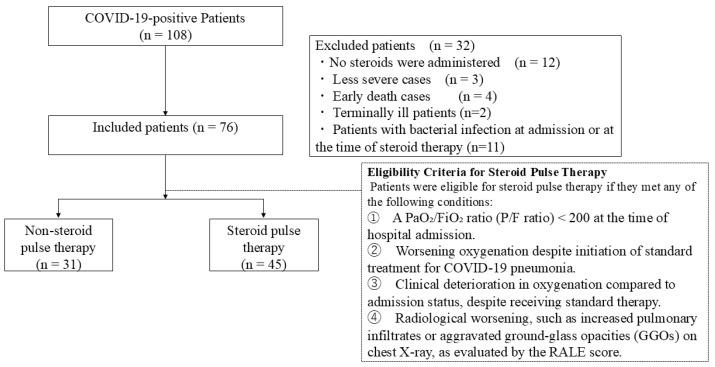
A flowchart illustrating the distribution of the study population. Abbreviations: COVID-19, coronavirus disease 2019; RALE, Radiographic Assessment of Lung Edema.

**Figure 2 viruses-17-00822-f002:**
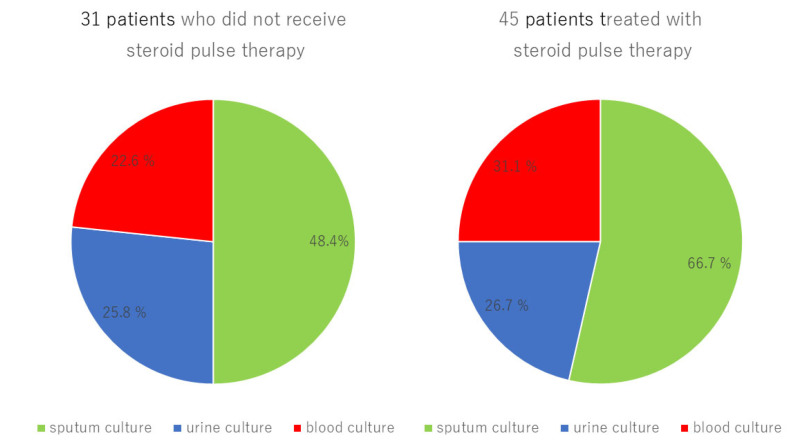
Comparison of secondary infections between the steroid pulse therapy and non-steroid pulse therapy groups.

**Table 1 viruses-17-00822-t001:** Comparison of patient characteristics in the steroid pulse therapy ** and non-steroid pulse therapy groups.

Factor	All (*n* = 76)	Steroid Pulse Therapy (*n* = 45)	Non-Steroid Pulse therapy (*n* = 31)	*p* Value *
(A) Parameters				
Age (years)	61.0 (54.8–75.0)	60.0 (54.0–75.0)	71.0 (57.0–76.5)	0.1443
Male, *n* (%)	58 (76.3)	36 (80)	22 (71)	0.3627
BMI (kg/m^2^)	25.7 (22.2–27.7)	26.3 (24.0–27.7)	23.1 (20.8–27.0)	0.043
sBP (mmHg)	130 (117–143)	130 (86–171)	132 (70–176)	0.959
HR (bpm)	82.5 (73.5–97.3)	84.0 ± 18.8	87.4 ± 17.8	0.408
RR	24.6 ± 6.9 (12–57)	24.2 ± 6.93 (16–57)	25.0 ± 6.86 (12–40)	0.521
P/F ratio	150.0 (90.9–271.4)	200.0 (76.0–316.7)	130.0 (97.0–220.0)	0.377
Comorbidities				
Diabetes (*n*, %)	28 (36.8)	18 (40.0)	10 (32.2)	0.492
Cancer (*n*,%)	6 (7.9)	3 (6.7)	3 (9.7)	0.632
Hypertension (*n*,%)	37 (48.7)	22 (48.9)	15 (48.4)	0.966
CRF (*n*, %)	2 (2.6)	0 (0)	2 (6.5)	0.084
Smoking (*n*,%)	40 (52.6)	25 (55.6)	15 (48.4)	0.539
Duration from onset to hospitalization (day)	6.0 (3.0–8.0)	6.0 (4.0–7.0)	6.0 (2.5–8.0)	0.924
Duration from onset to steroid administration (days)	6.0 (4.0–8.0)	6 (0–14)	6 (0–30)	0.836
(B) Blood biochemical examination				
WBC (×10^3^/μL)	6.9 (5.1–9.4)	6.4 (4.6–8.6)	8.1 (5.4–10.3)	0.158
Lymphocyte (×10^3^/μL)	0.7 (0.4–0.9)	0.6 (0.4–0.9)	0.8 (0.6–1.0)	0.045
Neutrophil (10^3^/μL)	5.9 (4.1–8.4)	5.6 (3.5–7.0)	7.0 (4.9–9.3)	0.021
Hgb, g/dL	13.7 (12–15.3)	13.8 (12.1–15.6)	12.9 (12.0–14.3)	0.100
Ht (%)	39.7 ± 7.48 (1.0–51.9)	41.0 (1–51.9)	39.3 (26.4–50.1)	0.139
Plt (×10^3^/μL)	183.0 ± 65.5 (38–410)	190.1 ± 68.1	190.6 ± 62.7	0.604
D-dimer (μg/mL)	1.2 (1.0–3.0)	1.2 (1.0–2.3)	1.4 (1.0–4.0)	0.377
LDH (U/L)	451.5 (338–596.5)	499.0 (338.0–611.0)	398.0 (327.0–573.5)	0.288
CK (U/L)	127.5 (55.8–435.8)	131 (61–410)	121 (38–639)	0.583
AST (U/L)	47.5 (31.8–76)	49 (36–76)	44 (31–69)	0.479
ALT (U/L)	31.5 (18.8–52.8)	36.0 (20.0–56.0)	25.0 (17.0–47.5)	0.267
CRP (mg/dL)	10.0 (5.4–17.7)	9.6 (5.8–15.9)	11.5 (4.9–21.6)	0.623
HbA1c (NGSP (%))	6.6 (6.1–7.8)	6.6 (6.1–7.8)	6.5 (6.1–8.0)	0.928
Ferritin (ng /mL)	697.5 (339.1–1397.3)	554.0 (326.0–1232.5)	882.0 (363.0–1421.0)	0.462
KL-6 (U/mL)	380 (219–667)	637.0 (344.5–1288.5)	511.5 (365.3–820.0)	0.728
Lactate (mmoL/L)	1.3 (1.0–1.6)	1.3 (1.1–2.1)	1.2 (0.8–1.6)	0.161
SOFA score	4 (3–6)	4.0 (3.0–5.0)	4.0 (3.0–6.5)	0.416
CONUT score	7 (5–9)	7.0 (5.0–9.0)	6.0 (5.5–9.0)	0.819
(C) Treatment				
Favipiravir, *n* (%)	24/76 (31.6)	11/45 (24.4)	13/31 (42.0)	0.107
Remdesivir, *n* (%)	547/7 (71.1)	36 (80.0)	18 (58.1)	0.038
Tocilizumab, *n* (%)	53/76 (69.7)	33/45 (73.3)	20/31 (64.5)	0.411
Baricitinib, *n* (%)	11/76 (14.5)	10/45 (22.2)	1/31 (3.22)	0.021
Anticoagulant therapy, *n* (%)	74/76 (97.4)	45/45 (100)	29/31 (93.5)	0.084
Steroid therapy, *n* (%)	68/76 (89.5)	37/45 (82.2)	31/31 (100)	0.013
(D) Respiratory severity				
Intubation, *n* (%)	40/76 (52.6)	27/45 (60.0)	13/31 (41.9)	0.121
VFD	1.5 (0–25.0)	0 (0–17)	15 (3–22)	0.0295
(E) Secondary infection				
Secondary infection, *n* (%) ***	55 (72.4)	34 (75.6)	17 (54.8)	0.0589

* Categorical variables are presented as numbers (*n*) and percentages (%), and nonparametric continuous variables as medians and interquartile ranges (IQRs; first quartile and third quartile). Continuous variables were compared using Student’s t-test or the Mann–Whitney U test, as appropriate. Chi-square tests or Fisher’s exact probability tests were performed for categorical variables. We determined the significance level to be 5%. ** Steroid therapy was defined as dexamethasone (6 mg/day) administered as standard steroid therapy or methylprednisolone (1000 mg/day) administered daily for three days. *** Secondary infections refer to all bacterial and fungal infections that occurred after the start of steroid therapy. Samples were collected from sites of possible infection and cultured. The cause of infection was determined by combining the clinical, imaging, and laboratory data, as well as the bacterial culture results. Abbreviations: BMI, body mass index; CRF, chronic renal failure; SOFA, sequential organ failure assessment; sBP, systolic blood pressure; HR, heart rate; RR, respiratory rate; P/F, partial pressure of arterial oxygen/fraction of inspired oxygen; WBC, white blood cells; Hgb, hemoglobin; Plt, platelets; LDH, lactate dehydrogenase; CK, creatine kinase; AST, aspartate aminotransferase; ALT, alanine aminotransferase; CRP, C-reactive protein; HbA1c, hemoglobin A 1c; NGSP, National Glycohemoglobin Standardization Program; KL-6, sialylated carbohydrate antigen; CONUT, controlling nutritional status; VFD, ventilation-free days.

**Table 2 viruses-17-00822-t002:** Multiple logistic regression analysis of factors associated with prognosis in severe COVID-19 pneumonia.

Factor	Correlation Co-Efficient	SE	Odds Ratio	*p*-Value	95% CI
Age	−0.010	0.003	0.920	0.005	0.868–0.974
Ferritin	-				
Lactate	-				
Smoking	-				
Fabipiravir	-				
Remdecivir	0.286	0.105	8.202	0.016	1.479–49.495
Tocilizumab	-				
Steroid pulse therapy	−0.412	0.094	0.032	<0.001	0.004–0.240

CI, confidence interval; COVID-19, coronavirus disease; SE, standard error.

**Table 3 viruses-17-00822-t003:** Multiple logistic regression analysis of prognostic factors associated with patients with severe COVID-19 pneumonia receiving steroid pulse therapy (*n* = 45).

Factor	Correlation Coefficient	SE	Odds Ratio	*p*-Value	95% CI
Age (years)	−0.209	0.074	0.811	0.005	0.702–0.938
Remdesivir, *n* (%)	3.650	1.729	38.49	0.035	1.30–1140.0

Variables included in the analysis included age, BMI, smoking, LDH, ferritin, KL-6, WBC, lactate, favipiravir, remdesivir, and duration from onset to hospitalization. BMI, body mass index; LDH, lactate dehydrogenase; KL-6, sialylated carbohydrate antigen; CI, confidence interval; COVID-19, coronavirus disease; WBC, white blood cells; SE, standard error. According to the predictive model equation ([Age] × −0.209 + [received remdesivir treatment] × 3.650 + 10.284), the target rate was 84.4%.

## Data Availability

Data supporting the findings of this study are available from the corresponding author, J.Y., on reasonable request.

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
