# Peer review of "Steroid Pulse Therapy Leads to Secondary Infections and Poor Outcomes in Patients with Severe Acute Respiratory Syndrome Coronavirus 2 (SARS-CoV-2) in Intensive Care Units: A Retrospective Cohort Study"

_viruses, 2025, doi:10.3390/v17060822_

Round 1

Reviewer 1 Report

Comments and Suggestions for Authors

I reviewed the viruses-3609616 manuscript. The authors retrospectively evaluated the effect of steroid pulse therapy in 76 pts with severe SARS-CoV-2 pneumonia. The authors demonstrated that steroid pulse therapy does not offer any additional advantage over the administration of low dose dexamethazone plus remdesivir. On the contrary, they found that steroid pulse therapy increases the risk of mortality in those pts. Although their findings is of considerable significance, the manuscript presents certain defects, which -in my opinion- make it not suitable for scientific publication (at least in its current form). The most important ones are the following:

1) The study lacks novelty.

2) The sample size is very small and all the participants come from a single institution.

3) The inclusion and exclusion criteria are not clearly defined. In this regard, it is not depicted in figure 1 (flow chart) the original number of pts, who were eligible at first. Secondly, the criterion for steroid pulse therapy is not clearly described (ie was it under the discretion of the supervising physician?). Third, it is not clear whether all pts included in the study suffered from disease deterioration, as it is not depicted whether P/F ratio and/or murray score got worse. In this regard, it is -in my opinion- not right to include only one parameter from the murray score (in this case imaging findings). Therefore, either murray score should be calculated again or be totally excluded.

4) Another source of bias is that all pts in the non-steroid pulse group received conventional treatment with dexamethazone, while in the steroid pulse group only a proportion of those received conventional treatment.

5) Comments on the multivariate analysis: 

  • Since a prospective validation cohort has been absent, the findings reflect associations rather than predictions.
  • Data from goodness-of-fit tests are lacking.
  • It is not clear how the different co-variates were selected for the final model. Additionally, the relatively high number of co-variates in comparison with the small sample size may be responsible for a certain degree of overfitting.
  • VIF numbers are high, reflecting a high degree of multicolinearity.
  • Analysis of data in different time points is not multivariate and differences may reflect the effect of other factors and this should be clearly pointed out in the "limitations" paragraph.

5) The authors should discuss their findings along with previous findings demonstrating that pts with severe COVID-19, who improved upon treatment with dexamethazone, exhibited different lymphocyte subpopulation in the peripheral blood:

(Pappas AG, Chaliasou AL, Panagopoulos A, Dede K, Daskalopoulou S, Moniem E, Polydora E, Grigoriou E, Psarra K, Tsirogianni A, Kalomenidis I. Kinetics of Immune Subsets in COVID-19 Patients Treated with Corticosteroids. Viruses. 2022 Dec 24;15(1):51. doi: 10.3390/v15010051.)

Minor comments:

1) P-value in SOFA score over time is <0.05, while in the text is described as non-significant.

2) The comments regarding table 2 in the "results" should be transferred to the 'discussion" section

3) In line 38 (introduction) it is stated that "studies from other countries have reported better outcomes (6,7)", while the references listed there report the exact opposite.

Reviewer 2 Report

Comments and Suggestions for Authors

the authors tried to evaluate the effectiveness of steroid pulse therapy for treating SARS-CoV-2 in ICU and its relationships with secondary infections. Patient data was selected and analyzed to try to gain conclusion. The results were basically sound but I found the title and the conclusion part of the article a little bit confusing. The title of this article is very clear and decisive, claiming the steroid pulse therapy lead to secondary infections and poor out comes; but in the conclusions this remains controversial.

Round 2

Reviewer 1 Report

Comments and Suggestions for Authors

The authors significantly improved the manuscript by responding to a point-by-point way to the issues raised during the first review round.